# REGULARIZED MUTUAL INFORMATION NEURAL ESTIMATION

## ABSTRACT

With the variational lower bound of mutual information (MI), the estimation of MI can be understood as an optimization task via stochastic gradient descent. In this work, we start by showing how Mutual Information Neural Estimator (MINE) searches for the optimal function $T$ that maximizes the Donsker-Varadhan representation. With our synthetic dataset, we directly observe the neural network outputs during the optimization to investigate why MINE succeeds or fails: We discover the drifting phenomenon, where the constant term of $T$ is shifting through the optimization process, and analyze the instability caused by the interaction between the $logsumexp$ and the insufficient batch size. Next, through theoretical and experimental evidence, we propose a novel lower bound that effectively regularizes the neural network to alleviate the problems of MINE. We also introduce an averaging strategy that produces an unbiased estimate by utilizing multiple batches to mitigate the batch size limitation. Finally, we show that $L^2$ regularization achieves significant improvements in both discrete and continuous settings.

## 1 INTRODUCTION

Identifying a relationship between two variables of interest is one of the great linchpins in mathematics, statistics, and machine learning (Goodfellow et al., 2014; Ren et al., 2015; He et al., 2016; Vaswani et al., 2017). Not surprisingly, this problem is closely tied to measuring the relationship between two variables. One of the fundamental approaches is information theory-based measurement, namely the estimation of mutual information (MI). Recently, Belghazi et al. (2018) proposed a neural network-based MI estimator, which is called Mutual Information Neural Estimator (MINE). Due to its differentiability and applicability, it motivated several researches such as various loss functions bridging the gap between latent variables and representations (Chen et al., 2016; Belghazi et al., 2018; Oord et al., 2018; Hjelm et al., 2019), and methodologies identifying the relationship between input, output and hidden variables (Tishby & Zaslavsky, 2015; Shwartz-Ziv & Tishby, 2017; Saxe et al., 2019). Although many have shown the computational tractability and its usefulness, many intriguing questions about the MI estimator itself remain unanswered.

- How does the neural network inside MINE behave when estimating MI?
- Why does MINE loss diverge in some cases? Where does the instability originate from?
- Can we make a more stable estimate on small batch size settings?
- Why does the value of each term in MINE loss are shifting even after the estimated MI converges? Are there any side effects of this phenomenon?

This study attempts to answer these questions by designing a synthetic dataset to interpret network outputs. Through keen observation, we dissect the Donsker-Varadhan representation (DV) one by one and conclude that the instability and the drifting are caused by the interrelationship between the stochastic gradient descent based optimization and the theoretical properties of DV. Based on these insights, we extend DV to draw out a novel lower bound for MI estimation, which mitigates the aforementioned problems, and circumvents the batch size limitation by maintaining the history of network outputs. We furthermore look into the $L^2$ regularizer form of our bound in detail and analyze how various hyper-parameters impact the estimation of MI and its dynamics during the optimization process. Finally, we demonstrate that our method, called ReMINE, performs favorably against other existing estimators in multiple settings.

## 2 RELATED WORKS

**Definition of Mutual Information**   The mutual information between two random variables $X$ and $Y$ is defined as

$$I(X;Y) = D_{KL}(\mathbb{P}_{XY} || \mathbb{P}_X \otimes \mathbb{P}_Y) = \mathbb{E}_{\mathbb{P}_{XY}}[\log \frac{d\mathbb{P}_{XY}}{d\mathbb{P}_{X \otimes Y}}] \tag{1}$$

where $\mathbb{P}_{XY}$ and $\mathbb{P}_X \otimes \mathbb{P}_Y$ are the joint and the marginal distribution, respectively. $D_{KL}$ is the Kullback-Leibler (KL) divergence. Without loss of generality, we consider $\mathbb{P}_{XY}$ and $\mathbb{P}_X \otimes \mathbb{P}_Y$ as being distributions on a compact domain $\Omega \subset \mathbb{R}^d$.

**Variational Mutual Information Estimation**   Recent works on MI estimation focus on training a neural network to represent a tight variational MI lower bound, where there are several types of representations. Although these methods are known to have statistical limitations (McAllester & Stratos, 2018), their versatility is widely employed nonetheless (Hjelm et al., 2019; Veličković et al., 2018; Polykovskiy et al., 2018; Ravanelli & Bengio, 2018; François-Lavet et al., 2019).

One of the most commonly used is the Donsker-Varadhan representation, which is first used in Belghazi et al. (2018) to estimate MI through neural networks.

**Lemma 1.** *(Donsker-Varadhan representation (DV))*

$$I(X;Y) = \sup_{T:\Omega \to \mathbb{R}} \mathbb{E}_{\mathbb{P}_{XY}}[T] - \log(\mathbb{E}_{\mathbb{P}_X \otimes \mathbb{P}_Y}[e^T]). \tag{2}$$

*where both the expectations $\mathbb{E}_{\mathbb{P}_{XY}}[T]$ and $\mathbb{E}_{\mathbb{P}_X \otimes \mathbb{P}_Y}[e^T]$ are finite.*

However, as the second term in Eq. (2) leads to biased gradient estimates with a limited number of samples, MINE uses exponential moving averages of mini-batches to alleviate this problem. To further improve the sampling efficiency of MINE, Lin et al. (2019) proposes DEMINE that partitions the samples into train and test sets.

Other representations based on f-measures are also proposed by Nguyen et al. (2010); Nowozin et al. (2016), which produce unbiased estimates and hence eliminating the need for additional techniques.

**Lemma 2.** *(Nguyen, Wainwright, and Jordan representation (NWJ))*

$$I(X;Y) = \sup_{T:\Omega \to \mathbb{R}} \mathbb{E}_{\mathbb{P}_{XY}}[T] - \mathbb{E}_{\mathbb{P}_X \otimes \mathbb{P}_Y}[e^{T-1}], \tag{3}$$

*where the bound is tight when $T = log(dP/dQ) + 1$.*

Nevertheless, if MI is too big, estimators exhibit large bias or variation (McAllester & Stratos, 2018; Song & Ermon, 2020). To balance in between, Poole et al. (2019) design a new estimator $I_\alpha$ that interpolates Contrastive Predictive Coding (Oord et al., 2018) and NWJ.

Yet, these methods concentrate on various stabilization techniques rather than revealing the dynamics inside the black box. In this paper, we focus on the DV representation and provide intuitive understandings of the inner mechanisms of neural network-based estimators. Based on the analysis, we introduce a new regularization term for MINE, which can effectively remedy its weaknesses theoretically and practically.

## 3 HOW DOES MINE ESTIMATE?

Before going any further, we first observe the statistics network output in MINE during the optimization process using our novel synthetic dataset, and identify and analyze the following phenomena:

- Drifting phenomenon (Fig. 1a), where estimates of $\mathbb{E}_{\mathbb{P}_{XY}}[T]$ and $\log(\mathbb{E}_{\mathbb{P}_X \otimes \mathbb{P}_Y}[e^T])$ drifts in parallel even after the MI estimate converges.
- Exploding network outputs (Fig. 1d), where smaller batch sizes cause the network outputs to explode, but training with larger batch size reduces the variance of MI estimates (Fig. 2a).
- Bimodal distribution of the outputs (Fig. 2b), where the network not only classifies input samples but also clusters the network outputs as the MI estimate converges.

Based on these observations, we analyze the inner workings of MINE, and understand how batch size affects MI estimation.

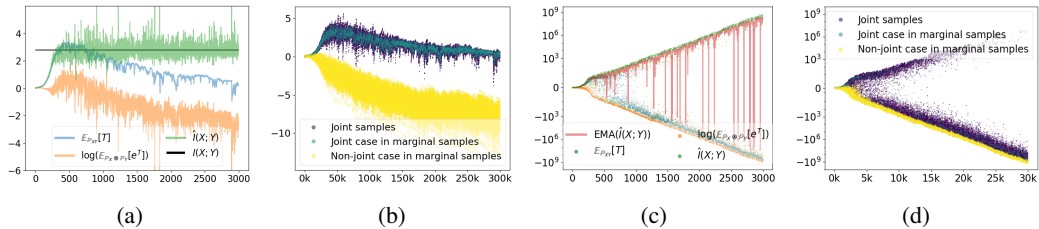

Figure 1: Training $T$ for 3000 iterations with batch size 100 [(a), (b)] and 10 [(c), (d)]. For (a) and (c), colored lines represent the estimation for $I(X;Y)$, $\mathbb{E}_{\mathbb{P}_{XY}}[T]$, and $\log(\mathbb{E}_{\mathbb{P}_X \otimes \mathbb{P}_Y}[e^T])$ of each batch. We also added the ideal MI for (a) and exponential moving average (EMA) of $\hat{I}(X;Y)$ with span 10 (decay rate $2/11$) for (c). We can see that MINE fails to generate a meaningful estimation on batch size 10. Outputs of the neural network for each sample are shown in (b) and (d). As there are 100 samples per iteration for (b) and 10 for (d), total number of outputs is 300000 and 30000, respectively. Note that the base-10 logarithmic scale is used on the y-axis for (c) and (d).

## 3.1 EXPERIMENT SETTINGS

**Dataset.** We designed a one-hot discrete dataset with uniform distribution $U(1, N)$ to estimate $I(X;X) = \log N$ with MINE, while easily discerning samples of joint distribution $X, X$ from marginal distribution $X \otimes X$. Additionally, we use one-hot representation to increase the input dimension, resulting in more network weights to train. In this paper, we used $N = 16$.

**Network settings.** We designed a simple statistics network $T$ with a concatenated vector of dimension $N \times 2 = 32$ as input. We pass the input through two fully connected layers with ReLU activation by widths: $32 - 256 - 1$. The last layer outputs a single scalar with no bias and activation. We used stochastic gradient descent (SGD) with learning rate $0.1$ to optimize the statistics network.

## 3.2 OBSERVATIONS

We can observe the drifting phenomenon in Fig. 1a, where the statistics of the network output are adrift even after the convergence of MINE loss. The analysis for this phenomenon will be covered in more detail with theoretical results in Section 4. This section will focus extensively on the relationship between batch size and $logsumexp$, and the classifying nature of MINE loss.

**Batch size limitation.** MINE estimates in a batch-wise manner, i.e., MINE uses samples inside a single batch when estimating $\mathbb{E}_{\mathbb{P}_{XY}}[T]$ and $\log(\mathbb{E}_{\mathbb{P}_X \otimes \mathbb{P}_Y}[e^T])$. Consider the empirical DV

$$\hat{I}(X;Y) = \sup_{T_\theta : \Omega \to \mathbb{R}} \mathbb{E}^{(n)}_{\widehat{\mathbb{P}}_{XY}}[T_\theta] - \log \mathbb{E}^{(n)}_{\widehat{\mathbb{P}_X \otimes \mathbb{P}_Y}}[e^{T_\theta}], \tag{4}$$

where $\mathbb{E}^{(n)}_{\widehat{\mathbb{P}}}$ is an empirical average associated with batch size $n$. Therefore, the variance of $\hat{I}(X;Y)$ increases as the batch size decreases. The observation in Fig. 2a is consistent with the batch size limitation problem (McAllester & Stratos, 2018; Song & Ermon, 2020), which shows that MINE must have a batch size proportional to the exponential of true MI to control the variance of the estimation.

**Exploding network outputs.** We can understand the output explosion problem in detail by comparing Fig. 1b and Fig. 1d. During optimization, network outputs of joint samples get increased by the first term of Eq. (4), where the inverse of the batch size is multiplied to the gradient of each network output. On the other hand, the output of marginal samples get decreased by the second term of Eq. (4) which concentrates the gradient to the maximum output. Note that the second term is dominated by the maximum network output due to $logsumexp$, which is a smooth approximation of $\max$. As a single batch is sampled from the true underlying distribution, joint case samples may or may not exist. If it exists, then the joint sample output dominates the term, and its output gets de-

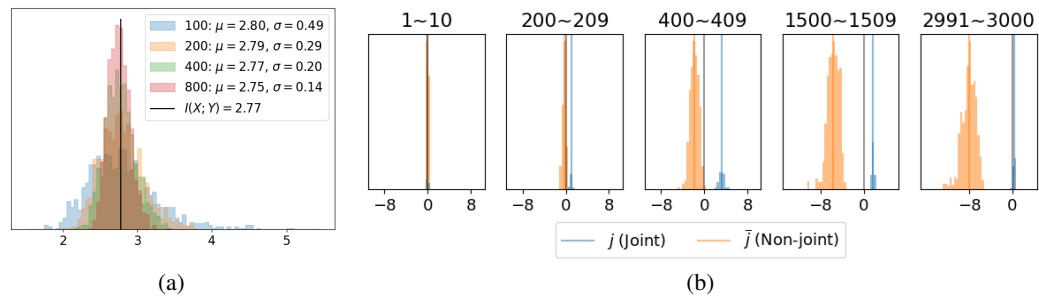

Figure 2: (a) Training $T$ with different batch sizes $100$, $200$, $400$, and $800$. We show the histogram of $\hat{I}(X;Y)$ of the final $1000$ batches. (b) Histogram of network outputs of marginal samples at different iterations. The probability of the joint case to occur is $\frac{1}{N}$, hence the proportionally small area of joint cases. A black vertical line is drawn at $0$ to assist visually.

creased accordingly, while other non-joint sample outputs also get slightly decreased. In summary, the second term acts as an occasional restriction for the increase of joint sample network outputs.[1]

The second term imposes a problem when the batch size is not large enough. With reduced sample size, joint samples that dominate the second term are getting rare. For the case where joint sample does not exist, marginal sample network outputs decrease much faster compared to the opposite case, and joint sample network outputs are more rarely restricted; thus network outputs diverge in both directions (Fig. 1d), and the second term vibrates between two extreme values depending on whether the joint case occurred (Fig. 1c). This obviously leads to numerical instability and estimation failure.

**Bimodal distribution of the outputs.** We furthermore observed network outputs directly, as both averaging terms of DV can inhibit the observation of how the statistics network acts on each sample. From the neural network viewpoint, whether each sample is realized from the joint or the marginal distribution is not distinguishable for the joint cases in marginal samples. Therefore, the statistics network has no means but to return the same output value, as it can be seen in Fig. 1b, indicating that the network can only separate joint and non-joint cases. This approach provides a clue that the network is solving a classification task, isolating joint samples from marginal samples, although the statistics network is only provided with samples from the joint and marginal distribution.

We observed the distribution of network outputs in detail, on the case where only the marginal samples are fed to the statistics network in Fig. 2b. It stands to reason that the network outputs follow a particular distribution, as the network output estimates a log-likelihood ratio between joint and marginal distribution with an added constant (Lemma 3). Through this, we can view the estimated MI as a sample average; hence Fig. 2a resembles the Gaussian noise by Central Limit Theorem (CLT).

Let us continue by concentrating on each network output. There is no distinction between the log-likelihood ratio of the samples in the same class for the one-hot discrete dataset: $j$ for the joint case and $\bar{j}$ for the non-joint case. This explains the classifying nature of the statistics network, and there have to be exactly two clusters in Fig. 2b. Also, as $\bar{j}$ becomes $-\infty$, $\exp(\bar{j})$ nears $0$, and $\exp(j)$ is few magnitudes bigger than $\exp(\bar{j})$ (see Fig. 2b). As mentioned above, few joint cases dominate the second term, so the second term becomes inherently noisier than the first term. Note that the effectiveness of conventional methods, such as applying exponential moving average to the second term (Belghazi et al., 2018) or clipping the network output values to restrict the magnitude of network outputs (Song & Ermon, 2020) can also be understood with the analysis above.

In addition, we cannot interpret the network outputs directly as the log-likelihood ratio due to un-regularized outputs, or the drifting problem. We will look into this fundamental limitation of MINE in more detail in the next section.

---

[1] Loosely speaking, the first term slowly increases a lot of joint samples network outputs, in contrary to the second term which quickly decreases a few joint sample network outputs.

## 4  THE PROPOSED METHOD: ReMINE

We look into the drifting problem of Fig. 1a in detail to introduce a novel loss with a regularizer that constrains the magnitude of network outputs and enables the use of network outputs from multiple batches. All proofs are provided in Appendix.

First, let us concentrate on the optimal function $T^*$, which can be directly drawn from the DV representation. We start with the results of Belghazi et al. (2018),

**Lemma 3.** *(Optimal Function $T^*$). Let $\mathbb{P}$ and $\mathbb{Q}$ be distributions on $\Omega \subset \mathbb{R}^d$. For some constant $C^* \in \mathbb{R}$, there exists an optimal function $T^* = \log \frac{d\mathbb{P}}{d\mathbb{Q}} + C^*$ such that $D_{KL}(\mathbb{P}||\mathbb{Q}) = \mathbb{E}_{\mathbb{P}}[T^*] - \log(\mathbb{E}_{\mathbb{Q}}[e^{T^*}])$.*

Note that Mukherjee et al. (2020) directly utilize this fact to model the statistics network. Let us extend the result above, and show that $C^*$ can be any real number and $T^*$ still be optimal.

**Lemma 4.** *(Family of Optimal Functions). For any constant $C \in \mathbb{R}$, $T = \log \frac{d\mathbb{P}}{d\mathbb{Q}} + C$ satisfies $D_{KL}(\mathbb{P}||\mathbb{Q}) = \mathbb{E}_{\mathbb{P}}[T] - \log(\mathbb{E}_{\mathbb{Q}}[e^T])$.*

This explains the drifting phenomenon we encountered in Fig. 1a. $C^*$ is drifting as there are no penalties on different $C^*$s. Drifting can be stopped by freezing the network weights (Lin et al., 2019), but is there a direct way to regularize $C$ so that the neural network can concentrate on finding a single solution, rather than a family of solutions?

**Theorem 5.** *(ReMINE Loss Function) Let $d$ be a distance function on $\mathbb{R}$. For any constant $C' \in \mathbb{R}$ and function $T : \Omega \to \mathbb{R}$,*

$$D_{KL}(\mathbb{P}||\mathbb{Q}) = \sup_{T:\Omega \to \mathbb{R}} \mathbb{E}_{\mathbb{P}}[T] - \log(\mathbb{E}_{\mathbb{Q}}[e^T]) - d(\log(\mathbb{E}_{\mathbb{Q}}[e^T]), C'),$$

Note that for the optimal $T^*$, $\mathbb{E}_{\mathbb{P}_{XY}}[T^*] = I(X;Y) + C'$ and $\log(\mathbb{E}_{\mathbb{P}_{X \otimes Y}}[e^{T^*}]) = C'$. Based on Theorem 5, we propose a novel loss function by adding a new term $d(\log(\mathbb{E}_{\mathbb{P}_X \otimes \mathbb{P}_Y}[e^T]), C')$ that regularizes the drifting of $C^*$. The details of the ReMINE algorithm is as follows.

---

**Algorithm 1:** ReMINE

---

$\theta \leftarrow$ Initialize network parameters, $K \leftarrow$ Moving average window size, $i \leftarrow 0$

**repeat**

   Draw $J$ samples from the joint distribution: $(\mathbf{x}_i^{(1)}, \mathbf{y}_i^{(1)}), \cdots (\mathbf{x}_i^{(J)}, \mathbf{y}_i^{(J)}) \sim \mathbb{P}_{XY}$

   Draw $M$ samples from the marginal distribution: $(\bar{\mathbf{x}}_i^{(1)}, \bar{\mathbf{y}}_i^{(1)}), \cdots (\bar{\mathbf{x}}_i^{(M)}, \bar{\mathbf{y}}_i^{(M)}) \sim \mathbb{P}_X \otimes \mathbb{P}_Y$

   Evaluate the lower-bound:

     $\hat{\mathbb{E}}_{\mathbb{P}_{XY}} \leftarrow \frac{1}{J} \sum_{j=1}^J T_\theta(\mathbf{x}_i^{(j)}, \mathbf{y}_i^{(j)}) \,, \hat{\mathbb{E}}_{\mathbb{P}_X \otimes \mathbb{P}_Y} \leftarrow \log(\frac{1}{M} \sum_{m=1}^M e^{T_\theta(\bar{\mathbf{x}}_i^{(m)}, \bar{\mathbf{y}}_i^{(m)})})$

     $\nu(\theta) \leftarrow \hat{\mathbb{E}}_{\mathbb{P}_{XY}} - \hat{\mathbb{E}}_{\mathbb{P}_X \otimes \mathbb{P}_Y} - d(\hat{\mathbb{E}}_{\mathbb{P}_X \otimes \mathbb{P}_Y}, C)$

   Update the statistics network parameters: $\theta \leftarrow \theta + \nabla_\theta \nu(\theta)$

   Estimate MI based on the last window $W = [\max(0, i - K + 1), \min(K, i)]$ of size $K$:

     $\hat{I}(X;Y) = \frac{1}{J \cdot |W|} \sum_{w \in W} \sum_{j=1}^J T_\theta(\mathbf{x}_w^{(j)}, \mathbf{y}_w^{(j)}) - \log(\frac{1}{M \cdot |W|} \sum_{w \in W} \sum_{m=1}^M e^{T_\theta(\bar{\mathbf{x}}_w^{(m)}, \bar{\mathbf{y}}_w^{(m)})})$

   Next iteration: $i \leftarrow i + 1$

**until** *convergence*;

---

ReMINE differs from other MINE-like methods that rely on single batch estimates, as ReMINE can utilize all the previous network outputs after the convergence of $\hat{I}(X;Y)$ to establish a final estimate. To demonstrate why these MINE-like methods cannot use the network outputs from multiple batches, we first describe two basic averaging strategies and then show that both methods produce a biased estimate when the statistics network $T$ is drifting.

**Theorem 6.** *(Estimation Bias caused by Drifting) The two averaging strategies below produce a biased MI estimate when the drifting problem occurs.*

   *1. Macro-averaging (similar to that of Poole et al. (2019)): Establish a single estimate through the average of estimated MI from each batch.*

   *2. Micro-averaging (our method): Calculate DV representation using the average of the each individual network outputs.*

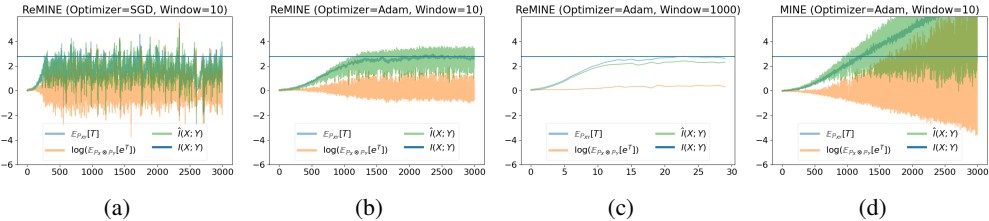

Figure 3: (a) Training $T$ with the same settings as Fig. 1d with ReMINE. We can see that ReMINE successfully avoids both drifting and network output explosion problem. (b) Same setting as (a), but used the optimizer from Section 5.2. We suspect Adam shows better performance as it accumulates previous gradients. (c) Same setting as (b) except we used micro-averaging strategy with sliding window of size 1000. (d) Training $T$ with the same settings as Fig. 1d, but used the same optimizer from (b). Outputs are more stabilized compared to Fig. 1d, but still fails to make a stable estimate.

## 5 EXPERIMENTAL RESULTS OF $L^2$ REGULARIZATION

Many choices exist for the distance function $d$ when implementing ReMINE. Here, we choose $d(x, C) = \lambda(x - C)^2$ and explain the rationale behind it. We also explore different choices of hyperparameters $C$ and $\lambda$. We show that ReMINE blocks the drifting effect by restricting $\log(\mathbb{E}_{\mathbb{P}_X \otimes \mathbb{P}_Y}[e^T])$ estimates to be $C$, and visualize the loss surface of ReMINE for different choices of $\lambda$s. Finally, we show that ReMINE achieves better performance than state-of-the-art methods in the continuous domain, and shows comparable performance on self-consistency tests.[2]

### 5.1 EFFECTIVENESS OF $L^2$ REGULARIZATION

For the sake of brevity, let batch size $B$, step size $\gamma$, $f = \mathbb{E}_{\widehat{\mathbb{P}}_{XY}}^{(n)}[T_\theta]$, $g = \log \mathbb{E}_{\widehat{\mathbb{P}_X \otimes \mathbb{P}_Y}}^{(n)}[e^{T_\theta}]$, and $C = 0$. Then, $\nabla \hat{I}(X; Y) = \nabla f - \nabla g - 2\lambda g \nabla g = \nabla f - \nabla g(1 + 2\lambda g)$. As previously discussed, $f$ increases each $B$ joint sample outputs with $(\gamma/B)\nabla T_\theta$. On the other hand, $g$ is the approximation of the maximum marginal sample output, so the gradient becomes close to $\gamma(1 + 2\lambda g)\nabla T_\theta$ for the maximum output. We can break down the dynamics of $g$ into two parts, depending on whether joint samples exist in $g$. If not, maximum marginal sample output gets reduced with step size $\gamma(1 + 2\lambda g)$, which is, unlike MINE, adaptively adjusted by the size of $g$. Hence, ReMINE regularizes the maximum output to be centered around $-\frac{1}{2\lambda}$, preventing the network outputs from diverging to $-\infty$. If any joint sample exists, its network output will be big enough to dominate $g$. Step size is also $\gamma(1 + 2\lambda g)$, which regularizes the joint sample network outputs more strongly as it increases. This, too, helps to avoid the output explosion in the $+\infty$ side, as shown in Fig. 3.

**Impact of $C$.** Fig. 4a shows the impact of changing $C$ on MI estimation, with the same settings as Fig. 1a. We observed that the newly added regularizer penalizes the $\log(\mathbb{E}_{\mathbb{P}_X \otimes \mathbb{P}_Y}[e^T])$ term to converge towards $C$ as expected, without losing the ability to estimate MI.

**Impact of $\lambda$.** As we observed in Section 3.2, the network outputs of joint and non-joint cases converge to $j$ and $\bar{j}$, respectively. Using this, we visualize the effect of the proposed method by drawing the loss surface of MINE and ReMINE for the one-hot discrete dataset in Fig. 5.

$$\mathcal{L}_{MINE} = \mathbb{E}_{\mathbb{P}_{XY}}[T] - \log(\mathbb{E}_{\mathbb{P}_X \otimes \mathbb{P}_Y}[e^T]) = j - \log(pe^j + (1 - p)e^{\bar{j}}) \tag{5}$$

$$\mathcal{L}_{ReMINE} = \mathbb{E}_{\mathbb{P}_{XY}}[T] - \log(\mathbb{E}_{\mathbb{P}_X \otimes \mathbb{P}_Y}[e^T]) - d(\log(\mathbb{E}_{\mathbb{P}_X \otimes \mathbb{P}_Y}[e^T]), C) \tag{6}$$

$$= j - \log(pe^j + (1 - p)e^{\bar{j}}) - \lambda(\log(pe^j + (1 - p)e^{\bar{j}}) - C)^2. \tag{7}$$

We again observe the drifting phenomena, as the loss surface has a plateau of equally palatable solutions. Regularization term successfully warps the loss surface, so that it has a single solution. As $\lambda$ increases, the loss surface becomes steeper, resulting in sporadic spikes for each gradient step.

---

[2]We release our code in

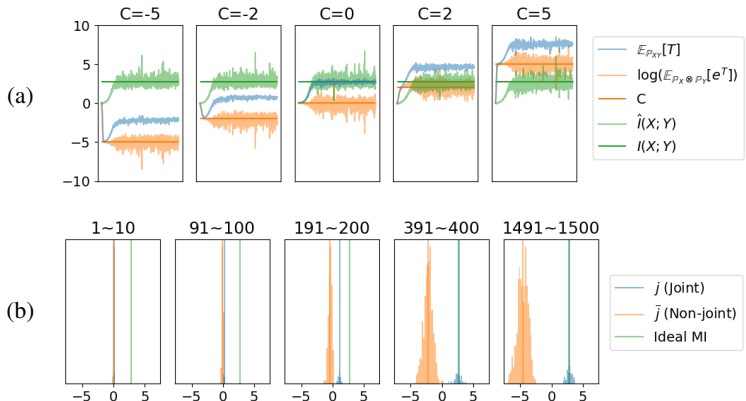

Figure 4: (a) Applying ReMINE for 1500 iterations, with different $C$s and $\lambda = 0.1$. (b) Histogram of network outputs for the marginal samples at different iterations. We set $C = 0$ and $\lambda = 0.1$. $\exp(j)$ and $\exp(\bar{j})$ converges to $N$ and $0$, respectively. These are the likelihood ratios between joint and marginal distribution for joint and non-joint cases. We can now directly interpret the network outputs, and $E(j)$ directly converges to ideal MI $\log N$, thanks to the regularization of $C$.

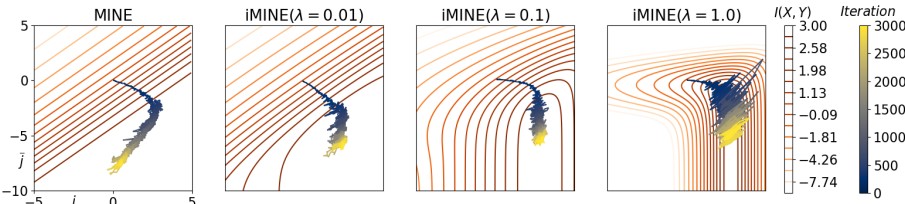

Figure 5: Comparing the loss surface for varying lambda. Per each batch, we averaged the network outputs of joint samples to estimate $j$, and non-joint cases of marginal samples to estimate $\bar{j}$.

## 5.2 ReMINE in the Continuous Domain

**20-D Gaussian dataset.** We sampled $(x, y)$ from $d$-dimensional correlated Gaussian dataset where $X \sim N(\mathbf{0}, \mathbf{I}_d)$ and $Y \sim N(\rho X, (1 - \rho^2)\mathbf{I}_d)$ given the correlation parameter $0 \leq \rho < 1$, which is taken from Belghazi et al. (2018) to test ReMINE on continuous random variables. The true MI for the dataset is $I(X, Y) = -\frac{d}{2}\log(1 - \rho^2)$. In these experiments, we set batch size to $64$.

**Network settings.** We consider a joint architecture, which concatenates the inputs $(x, y)$, and then passes through three fully connected layers with ReLU activation (excluding the output layer) by widths $40 - 256 - 256 - 1$, same as the network used in Poole et al. (2019). We used Adam optimizer with learning rate $5 \times 10^{-4}$, $\beta_1 = 0.9$ and $\beta_2 = 0.999$.

**Comparison to state-of-the-arts.** As mentioned in Fig. 4b, we can remove the $\log(\mathbb{E}_{\mathbb{P}_X \otimes \mathbb{P}_Y}[e^T])$ term by choosing $C = 0$. As discussed in Section 3.2, the second term is inherently noisy. Hence, using all the terms in ReMINE only in optimization and removing the second term in estimation can effectively reduce noise. We call this trick ReMINE-J.

To verify the quality of lower bounds, we compare ReMINE and ReMINE-J to InfoNCE (Oord et al., 2018), JS (Hjelm et al., 2019; Poole et al., 2019), MINE (Belghazi et al., 2018), NWJ (Nguyen et al., 2010), SMILE (Song & Ermon, 2020), SMILE+JS (which estimates with SMILE, and optimizes with JS), TUBA (Barber & Agakov, 2004; Poole et al., 2019) and $I_\alpha$ (Poole et al., 2019). To make a fair comparison, ReMINE also uses the macro-averaging strategy, the same as the other methods. Our methods show comparable or better estimation performance with less variance than others, as shown in Fig. 6. Exact values for bias, variance, and mean square error to the true MI for each estimator are shown in Appendix.

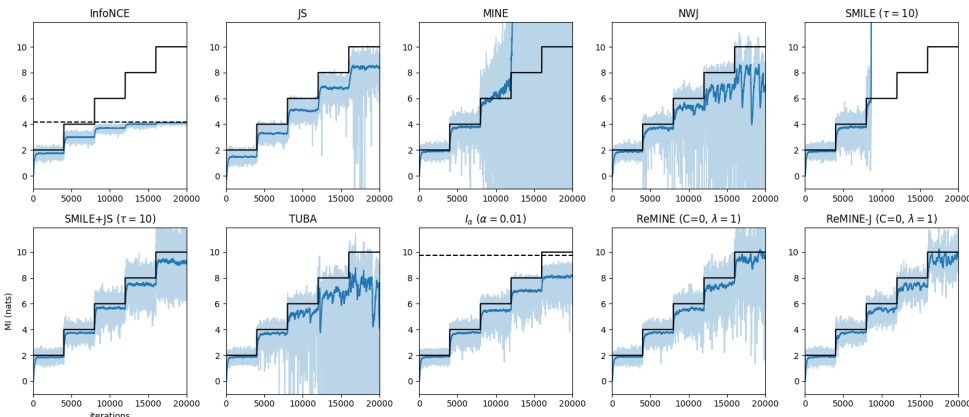

Figure 6: Estimation performance on 20-D Gaussian. Similar to Poole et al. (2019), we increase $\rho$ every $4000$ iterations. The estimated MI (light) and smoothed estimation with exponential moving average (dark) are plotted for each methods, and theoretical bounds are plotted by dotted lines.

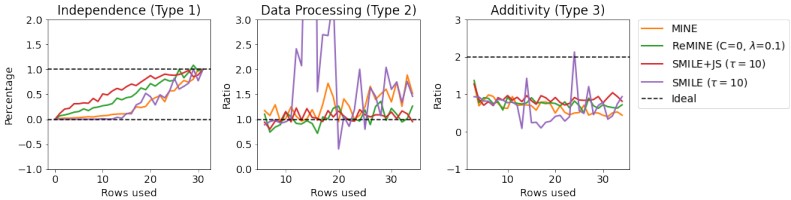

Figure 7: Self-consistency tests on CIFAR-10 (Krizhevsky et al., 2009). We report the average result of 10 repeated runs. Dotted lines indicate theoretical bounds for type 1, and ideal ratio for type 2, 3.

### 5.3 ReMINE in the Image Domain

**Experiment settings.** Song & Ermon (2020) introduced the self-consistency tests on image datasets to verify whether the MI estimates follow the basic properties of MI. For our experiments, we used the same network and optimizer, types of tests, and the number of epochs of Song & Ermon (2020) with batch size 16. We did not use our micro-averaging strategy for a fair comparison.

**Comparison on self-consistency tests.** To compare the stability of DV-based estimators, we conducted self-consistency tests on ReMINE, MINE, SMILE, and SMILE+JS. For type 1, every estimator successfully returns values between the theoretical bound with an increasing trend. For type 2, only ReMINE and SMILE+JS estimates are close to the ideal value. For type 3, none of the estimators worked well. However, ReMINE shows smaller variance compared to MINE and has similar stability to SMILE+JS.

## 6 Conclusion

In this paper, we studied how the neural network inside MINE handles the MI estimation problem. We delved into the drifting problem, where two terms of DV continue to fluctuate together even after the MI estimate converges, and the explosion problem, where the network outputs become unstable due to properties of the second term in DV when batch size is small. Based on the analysis, we penalized the objective function for obtaining a unique solution by using $L^2$ regularization. Despite the simplicity, the proposed loss and the micro-averaging strategy mitigate drifting, exploding, and batch size limitation problems. Further, ReMINE enables us to directly interpret the network output values as the log-likelihood ratio of joint and marginal distribution probability and performs favorably against state-of-the-art methods. However, further investigation needs to be done on the impact of optimizers on the batch size limitation, and why DV-based estimators fail in some of the self consistency tests.

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

# 7 APPENDIX: PROOFS

## 7.1 PROOF OF FAMILY OF OPTIMAL FUNCTIONS

**Theorem.** *For any constant $C \in \mathbb{R}$, $T = \log \frac{d\mathbb{P}}{d\mathbb{Q}} + C$ satisfies $D_{KL}(\mathbb{P}||\mathbb{Q}) = \mathbb{E}_{\mathbb{P}}[T] - \log(\mathbb{E}_{\mathbb{Q}}[e^T])$.*

*Proof.* Suppose that $T = \log \frac{d\mathbb{P}}{d\mathbb{Q}} + C$. We can write the function $T = (T^* - C^*) + C$ by Lemma 3 in the manuscript.
Therefore,

$$\mathbb{E}_{\mathbb{P}}[T] = \mathbb{E}_{\mathbb{P}}[T^* - C^* + C]$$
$$= \mathbb{E}_{\mathbb{P}}[T^*] - C^* + C,$$

and

$$\log(\mathbb{E}_{\mathbb{Q}}[e^T]) = \log(\mathbb{E}_{\mathbb{Q}}[e^{T^* - C^* + C}])$$
$$= \log(e^{C - C^*} \mathbb{E}_{\mathbb{Q}}[e^{T^*}])$$
$$= (C - C^*) + \log(\mathbb{E}_{\mathbb{Q}}[e^{T^*}]).$$

Since $\mathbb{E}_{\mathbb{P}}[T] - \log(\mathbb{E}_{\mathbb{Q}}[e^T]) = \mathbb{E}_{\mathbb{P}}[T^*] - \log(\mathbb{E}_{\mathbb{Q}}[e^{T^*}])$, the function $T$ also optimal. □

## 7.2 PROOF OF ReMINE LOSS FUNCTION

**Theorem.** *Let $d$ be a distance function on $\mathbb{R}$. For any constant $C' \in \mathbb{R}$ and function $T : \Omega \to \mathbb{R}$,*

$$D_{KL}(\mathbb{P}||\mathbb{Q}) = \sup_{T:\Omega \to \mathbb{R}} \mathbb{E}_{\mathbb{P}}[T] - \log(\mathbb{E}_{\mathbb{Q}}[e^T]) - d(\log(\mathbb{E}_{\mathbb{Q}}[e^T]), C'),$$

*Proof.* i) For any $T$,

$$\mathbb{E}_{\mathbb{P}}[T] - \log(\mathbb{E}_{\mathbb{Q}}[e^T]) - d(\log(\mathbb{E}_{\mathbb{Q}}[e^T]), C') \le \mathbb{E}_{\mathbb{P}}[T] - \log(\mathbb{E}_{\mathbb{Q}}[e^T]).$$

Therefore, $\sup_{T:\Omega \to \mathbb{R}} \mathbb{E}_{\mathbb{P}}[T] - \log(\mathbb{E}_{\mathbb{Q}}[e^T]) - d(\log(\mathbb{E}_{\mathbb{Q}}[e^T]), C') \le D_{KL}(\mathbb{P}||\mathbb{Q})$.

ii) By the theorem above, there exists $T^* = \log \frac{d\mathbb{P}}{d\mathbb{Q}} + C'$ such that

$$D_{KL}(\mathbb{P}||\mathbb{Q}) = \mathbb{E}_{\mathbb{P}}[T^*] - \log(\mathbb{E}_{\mathbb{Q}}[e^{T^*}])$$

and

$$\log(\mathbb{E}_{\mathbb{Q}}[e^{T^*}]) = \log(\mathbb{E}_{\mathbb{Q}}[e^{C'} \frac{d\mathbb{P}}{d\mathbb{Q}}]) = \log(\int e^{C'} \frac{d\mathbb{P}}{d\mathbb{Q}} d\mathbb{Q}) = C'.$$

Therefore,

$$\sup_{T:\Omega \to \mathbb{R}} \mathbb{E}_{\mathbb{P}}[T] - \log(\mathbb{E}_{\mathbb{Q}}[e^T]) - d(\log(\mathbb{E}_{\mathbb{Q}}[e^T]), C') \ge \mathbb{E}_{\mathbb{P}}[T^*] - \log(\mathbb{E}_{\mathbb{Q}}[e^{T^*}]) - d(\log(\mathbb{E}_{\mathbb{Q}}[e^{T^*}]), C')$$
$$= D_{KL}(\mathbb{P}||\mathbb{Q})$$

Combining i) and ii) finishes the proof. □

### 7.3 PROOF OF ESTIMATION BIAS CAUSED BY DRIFTING

**Theorem.** *(Estimation Bias caused by Drifting) The two averaging strategies below produce a biased MI estimate when the drifting problem occurs.*

1. *Macro-averaging (similar to that of Poole et al. (2019)): Establish a single estimate through the average of estimated MI from each batch.*

2. *Micro-averaging (our method): Calculate DV representation using the average of the each individual network outputs.*

*Proof.* Let the outputs of $i$th batch, $j$th sample inside the batch as $T_{ij}^{(J)}$, $T_{ij}^{(M)}$, joint and marginal case respectively, and the output without drifting as $T_{ij}^*$, and drifting constant for each batch $C_i$. Then, $T_{ij} = T_{ij}^* + C_i$.

When the number of batch is $B$ and each batch size is $N$,

1. Macro averaging:

$$\frac{1}{B}\Sigma_i[\frac{1}{N}\Sigma_j T_{ij}^{(J)} - \log(\frac{1}{N}\Sigma_j \exp T_{ij}^{(M)})] \tag{8}$$

$$=\frac{1}{B}\Sigma_i[\frac{1}{N}\Sigma_j(T_{ij}^{(J*)} + C_i) - \log(\frac{1}{N}\Sigma_j \exp(T_{ij}^{(M*)} + C_i))] \tag{9}$$

$$=\frac{1}{B}\Sigma_i[\frac{1}{N}\Sigma_j(T_{ij}^{(J*)} + C_i) - \log(\frac{1}{N}\exp C_i\Sigma_j \exp T_{ij}^{(M*)})] \tag{10}$$

$$=\frac{1}{B}\Sigma_i[\frac{1}{N}\Sigma_j T_{ij}^{(J*)} - \log(\exp(-C_i)\frac{1}{N}\exp C_i\Sigma_j \exp T_{ij}^{(M*)})] \tag{11}$$

$$=\frac{1}{B}\Sigma_i[\frac{1}{N}\Sigma_j T_{ij}^{(J*)} - \log(\frac{1}{N}\Sigma_j \exp T_{ij}^{(M*)})] \tag{12}$$

$$=\frac{1}{NB}\Sigma_{ij}T_{ij}^{(J*)} - \frac{1}{B}\Sigma_i[\log(\frac{1}{N}\Sigma_j \exp T_{ij}^{(M*)})] \tag{13}$$

$$\neq\frac{1}{NB}\Sigma_{ij}T_{ij}^{(J*)} - \log(\frac{1}{NB}\Sigma_{ij} \exp T_{ij}^{(M*)}) \tag{14}$$

2. Micro averaging:

$$\frac{1}{NB}\Sigma_{ij}T_{ij}^{(J)} - \log(\frac{1}{NB}\Sigma_{ij} \exp T_{ij}^{(M)}) \tag{15}$$

$$=\frac{1}{NB}\Sigma_{ij}(T_{ij}^{(J*)} + C_i) - \log(\frac{1}{NB}\Sigma_{ij} \exp(T_{ij}^{(M*)} + C_i))) \tag{16}$$

$$=\frac{1}{NB}\Sigma_{ij}T_{ij}^{(J*)} - \log[(\frac{1}{NB}\Sigma_{ij} \exp(T_{ij}^{(M*)} + C_i))^{\frac{1}{B}\Sigma_i C_i}] \tag{17}$$

$$\neq\frac{1}{NB}\Sigma_{ij}T_{ij}^{(J*)} - \log(\frac{1}{NB}\Sigma_{ij} \exp T_{ij}^{(M*)}) \tag{18}$$

$\square$

## 8   APPENDIX: ADDITIONAL EXPLANATIONS

**Additional explanations for Fig. 4b**   As the $N$-dimensional one-hot discrete dataset is uniform, we can easily calculate the likelihood ratio of joint and non-joint case samples. For all the possible samples, $\mathbb{P}_X \otimes \mathbb{P}_Y = 1/N^2$, as they are total of $N^2$. Also, for joint case samples, $\mathbb{P}_{XY} = 1/N$, and $\mathbb{P}_{XY} = 0$ for non-joint case samples. Hence, the likelihood ratio for the joint cases is $N$, and non-joint cases is 0. These are consistent with the experimental results, where $j$ converges to $\log N$, and $\bar{j}$ keeps decreasing. Nonetheless, as $exp(\bar{j})$ gets closer to zero, the second term of ReMINE loss has lesser influence; hence the decreasing speed of $\bar{j}$ gets slowed down to a halt as it reaches $-\frac{1}{2\lambda} = -5$.

We can explain the same result from the perspective of $j$ and $\bar{j}$. As we observed in Section 3.2, the network output values of joint and non-joint cases converge to $j$ and $\bar{j}$, respectively. Since the dataset is uniform, the probability $p$ of joint cases appearing from the marginal samples is $\frac{1}{N}$. Therefore, we can analyze the value of $j$ and $\bar{j}$ after convergence as follows: as iteration $i \to \infty$,

$$\mathbb{E}_{\mathbb{P}_{XY}}[T(i)] = j \to I(X;Y) + C = \log N + C \tag{19}$$

$$\log(\mathbb{E}_{\mathbb{P}_X \otimes \mathbb{P}_Y}[e^{T(i)}]) = \log(pe^j + (1-p)e^{\bar{j}}) = \log(\frac{1}{N}e^j + \frac{N-1}{N}e^{\bar{j}}) \to C \tag{20}$$

where $T(i)$ is the statistics network at iteration $i$. We combine Eq. (19) and Eq. (20) to

$$\frac{1}{N}e^{\log N + C} + \frac{N-1}{N}e^{\bar{j}} \to e^C, \text{ and} \tag{21}$$

$$e^{\bar{j}} \to 0. \tag{22}$$

In summary, $j$ will converge to $\log N + C$, and $e^{\bar{j}}$ to 0, as shown in Fig. 4b. Note that $j$ and $\bar{j}$ serves as a back-of-the-envelope calculation for us to estimate network outputs easily on discrete settings.

**What happens if the batch size is small?**   When the batch size is 1 and $C = 0$, the loss function of ReMINE changes its characteristics as follows.

- **Joint case occurs.** As the samples are indistinguishable,

$$\mathcal{L} = j - \log e^j - d(j, 0) = -\lambda j^2, \tag{23}$$

   which is maximized when $j = 0$.
- **Non-joint case occurs.**

$$\mathcal{L} = j - \log e^{\bar{j}} + d(\log e^{\bar{j}}, 0) = j - \bar{j} - \lambda \bar{j}^2. \tag{24}$$

   The latter quadratic term of $\bar{j}$ is maximized when $\bar{j} = -\frac{1}{2\lambda}$.

If the statistics network succeeds to converge on both cases for our one-hot discrete dataset,

$$\hat{I}(X;Y) = \mathbb{E}^{(n)}_{\widehat{\mathbb{P}_{XY}}}[T_\theta] - \log \mathbb{E}^{(n)}_{\widehat{\mathbb{P}_X \otimes \mathbb{P}_Y}}[e^{T_\theta}] = 0 - \log(pe^0 + (1-p)e^{-\frac{1}{2\lambda}}) \to -\log p \tag{25}$$

when $\lambda \to +0$. As $p = \frac{1}{N}$, $\hat{I}(X;Y) \to \log N$.

Intuitively, on smaller batch sizes, joint cases cannot occur in marginal samples, as mentioned in Section 3.2. Hence, $\mathbb{E}_{\mathbb{P}_{XY}}[T]$ and $\log(\mathbb{E}_{\mathbb{P}_X \otimes \mathbb{P}_Y}[e^T])$ behave differently compared to the larger batch size. The regularizer term penalizes both terms in different ways. Joint cases in marginal samples can contribute only with Eq. (23), so $\mathbb{E}_{\mathbb{P}_{XY}}[T] \to 0$. Moreover, as $\lambda$ gets smaller, $\log(\mathbb{E}_{\mathbb{P}_X \otimes \mathbb{P}_Y}[e^T])$ gets regularized less so that it can converge to $-\hat{I}(X;Y)$. In contrast, since MINE has no regularization term, namely $\lambda = 0$, there is no way for the joint case in marginal samples to influence $T$, hence failing to estimate MI as shown in Fig. 8b.

**Impact of $\lambda$ with batch size.**   We inspect the relationship between batch size and $\lambda$ in detail. Fig. 8 shows that imposing regularization reduces noise on a large batch size domain. However, on a small batch size domain, $\log(\mathbb{E}_{\mathbb{P}_X \otimes \mathbb{P}_Y}[e^T])$ cannot have nonzero value, hence failing to estimate MI value. The effect of the ReMINE loss in two different domains gets mixed in between.

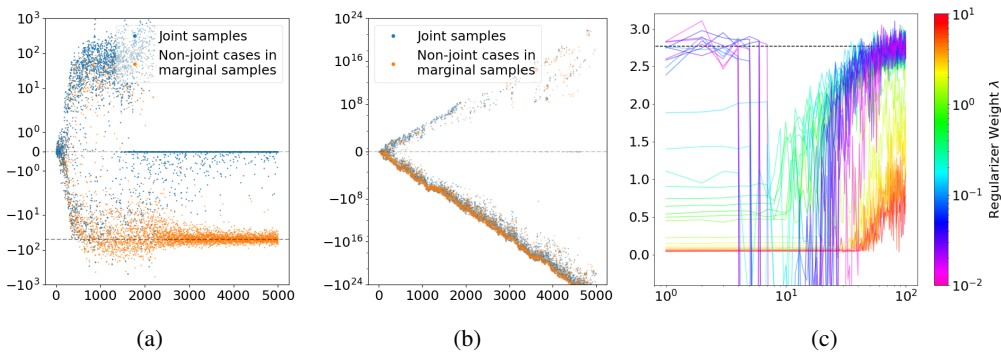

Figure 8: (a) Comparing the joint and non-joint case outputs of ReMINE with $\lambda = 0.01$ when batch size is 1 for 5000 iterations. We can see $j \to 0$ and $\bar{j} \to -\frac{1}{2\lambda} = -50$. (b) Comparing the joint and non-joint case outputs of MINE for 5000 iterations. As the statistics network struggles to diverge into two different values, it becomes numerically unstable, hence failing in the middle of the training. (c) Comparing the estimation performance of different $\lambda$s on varying batch sizes $1, 2, \cdots 100$. The dotted line represents the true MI.

**Visualizing network outputs on 1-D Gaussian.** The dataset forbids us to label joint and non-joint samples explicitly, so we visualized the network outputs on 2-D plane. We used the same experiment settings as Section 3.1, only changing the input dimension to 2. We can see in Fig. 9 that the network outputs of the overlapping region remain near 0, which indicates that the likelihood is equal between joint and marginal distribution. Other regions are separated by the sign of their outputs. Positive network output means that joint distribution is more probable than marginal distribution to sample that data point, and vice versa.

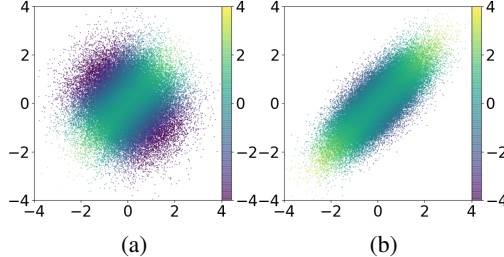

Figure 9: Visualizing network outputs on the 1-D correlated Gaussian dataset. Each axis represents a paired value $x$ and $y$ of each sample, and the color represents network outputs for (a) marginal and (b) joint samples.

## 9 APPENDIX: ADDITIONAL EXPERIMENTAL DETAILS

**Additional experiments outputs on** 20**-D Gaussian.** For quantitative comparison with other approaches on the 20-D correlated Gaussian dataset, we show the bias, variance and mean squared error (MSE) of the neural network-based and nearest neighbor-based MI estimators in Fig. 10, Fig. 11 and Table 1. We omitted values which are more than 100 in Table 1. We additionally show results from KL (Jiao et al., 2018), Mixed KSG (Gao et al., 2017), CCMI (Mukherjee et al., 2020), TNCE (Oord et al., 2018; Poole et al., 2019) and ReMINE-L1 (our method with L1 regularization).

Both ReMINE and ReMINE-J shows comparable or better performance compared to other methods. Note that L1 regularizer also suffers from the explosion problem, as the gradient is not adaptively adjusted by the magnitude of network outputs, as discussed in Section 5.1.

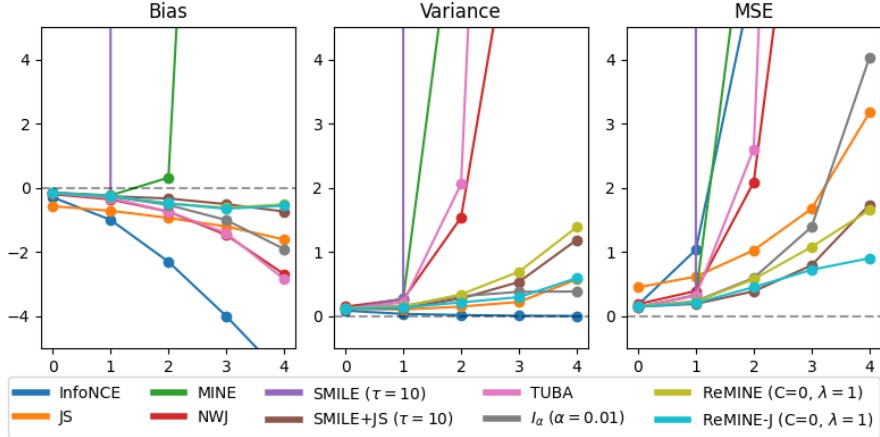

Figure 10: Bias, variance, MSE of estimators on 20-D correlated Gaussian dataset

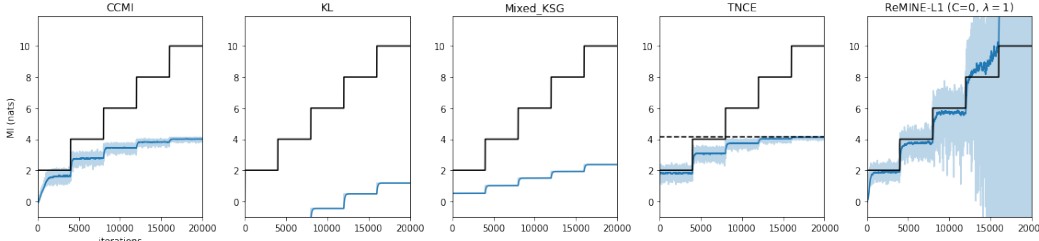

Figure 11: Estimation performance on 20-D correlated Gaussian dataset for additional estimators

| | MI | 2 | 4 | 6 | 8 | 10 |
|---|---|---|---|---|---|---|
| | CCMI | -0.65 | -1.24 | -2.55 | -4.18 | -6.00 |
| | InfoNCE | -0.30 | -1.00 | -2.31 | -4.00 | -5.89 |
| | JS | -0.57 | -0.72 | -0.94 | -1.21 | -1.61 |
| | KL | -4.35 | -5.25 | -6.46 | -7.52 | -8.83 |
| Bias | MINE | -0.15 | -0.24 | 0.32 | 33.81 | |
| | Mixed KSG | -1.49 | -2.99 | -4.51 | -6.08 | -7.64 |
| | NWJ | -0.20 | -0.36 | -0.74 | -1.47 | -2.70 |
| | SMILE ($\tau = 10$) | -0.15 | -0.26 | | | |
| | SMILE+JS ($\tau = 10$) | -0.19 | -0.26 | -0.34 | -0.51 | -0.74 |
| | TNCE | -0.19 | -0.93 | -2.26 | -3.98 | -5.88 |
| | TUBA | -0.17 | -0.33 | -0.74 | -1.42 | -2.85 |
| | $I_\alpha$ ($\alpha = 0.01$) | -0.16 | -0.28 | -0.53 | -1.01 | -1.91 |
| | ReMINE (C=0, $\lambda = 1$) | -0.16 | -0.27 | -0.49 | -0.62 | -0.51 |
| | ReMINE-L1 (C=0, $\lambda = 1$) | -0.25 | -0.27 | -0.33 | 0.78 | 40.44 |
| | ReMINE-J (C=0, $\lambda = 1$) | -0.16 | -0.27 | -0.49 | -0.65 | -0.56 |
| | CCMI | 0.24 | 0.03 | 0.02 | 0.01 | 0.00 |
| | InfoNCE | 0.09 | 0.03 | 0.02 | 0.01 | 0.00 |
| | JS | 0.12 | 0.10 | 0.15 | 0.22 | 0.57 |
| | KL | 0.00 | 0.00 | 0.00 | 0.00 | 0.00 |
| | MINE | 0.12 | 0.27 | 7.16 | | |
| Variance | Mixed KSG | 0.00 | 0.00 | 0.00 | 0.00 | 0.00 |
| | NWJ | 0.15 | 0.26 | 1.54 | 6.85 | 23.42 |
| | SMILE ($\tau = 10$) | 0.12 | 0.26 | | | |
| | SMILE+JS ($\tau = 10$) | 0.11 | 0.12 | 0.27 | 0.53 | 1.19 |
| | TNCE | 0.04 | 0.03 | 0.02 | 0.01 | 0.00 |
| | TUBA | 0.12 | 0.21 | 2.06 | 21.97 | 35.54 |
| | $I_\alpha$ ($\alpha = 0.01$) | 0.12 | 0.16 | 0.31 | 0.38 | 0.38 |
| | ReMINE (C=0, $\lambda = 1$) | 0.13 | 0.15 | 0.34 | 0.69 | 1.39 |
| | ReMINE-L1 (C=0, $\lambda = 1$) | 0.20 | 0.14 | 0.48 | 4.44 | |
| | ReMINE-J (C=0, $\lambda = 1$) | 0.12 | 0.13 | 0.21 | 0.29 | 0.59 |
| | CCMI | 0.67 | 1.57 | 6.54 | 17.51 | 35.95 |
| | InfoNCE | 0.17 | 1.03 | 5.33 | 16.02 | 34.68 |
| | JS | 0.45 | 0.62 | 1.03 | 1.67 | 3.17 |
| | KL | 18.92 | 27.54 | 41.71 | 56.56 | 77.94 |
| MSE | MINE | 0.14 | 0.33 | 7.27 | | |
| | Mixed KSG | 2.22 | 8.93 | 20.34 | 36.98 | 58.37 |
| | NWJ | 0.19 | 0.39 | 2.09 | 9.01 | 30.68 |
| | SMILE ($\tau = 10$) | 0.14 | 0.33 | | | |
| | SMILE+JS ($\tau = 10$) | 0.15 | 0.19 | 0.39 | 0.79 | 1.74 |
| | TNCE | 0.08 | 0.89 | 5.14 | 15.82 | 34.55 |
| | TUBA | 0.14 | 0.32 | 2.60 | 23.97 | 43.67 |
| | $I_\alpha$ ($\alpha = 0.01$) | 0.14 | 0.23 | 0.60 | 1.39 | 4.03 |
| | ReMINE (C=0, $\lambda = 1$) | 0.15 | 0.22 | 0.58 | 1.08 | 1.65 |
| | ReMINE-L1 (C=0, $\lambda = 1$) | 0.27 | 0.21 | 0.59 | 5.06 | |
| | ReMINE-J (C=0, $\lambda = 1$) | 0.15 | 0.20 | 0.45 | 0.72 | 0.90 |

Table 1: Bias, variance and MSE of estimators on 20-D correlated Gaussian dataset

**Additional experiments on the self consistency test.** We report the performance of other variational bound methods (JS, $I_\alpha$, InfoNCE) in Fig. 12. $I_\alpha$ and JS often result in unstable MI estimates, as shown in the type 2 experiment. On the other hand, InfoNCE estimates MI quite reliably but also fails for type 3.

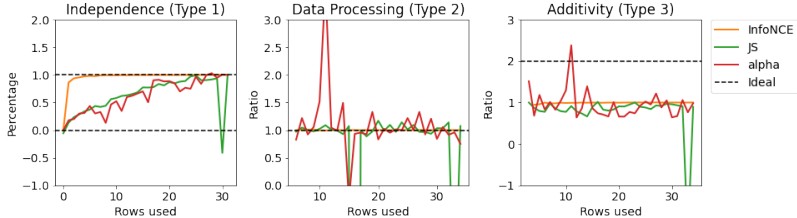

Figure 12: Self-consistency tests of JS, $I_\alpha$, and InfoNCE on CIFAR-10. We use the average result of 10 repeated runs. Dotted lines indicate theoretical bounds for type 1, and ideal ratio for type 2, 3.

To observe on a different dataset setting, we also used MNIST (LeCun et al., 1998). As shown in Fig. 13, tests yielded similar results.

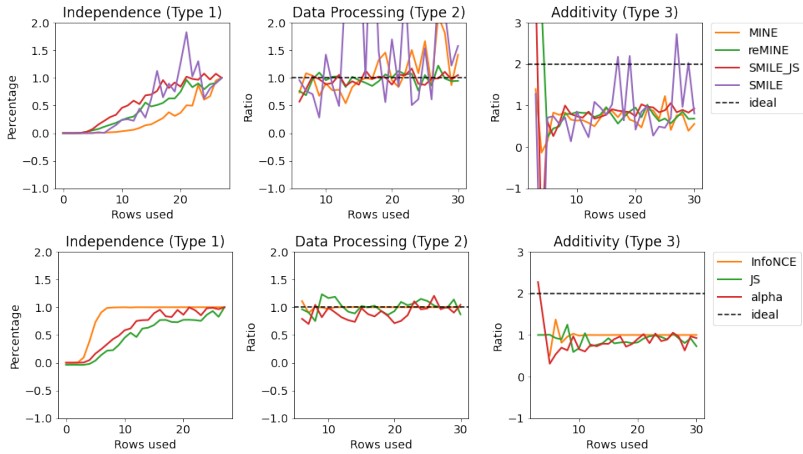

Figure 13: Self-consistency tests on MNIST.

To observe on a different statistics network, we used modified ResNet18 (He et al., 2016) that outputs a single scalar. As shown in Fig. 14, SMILE has become more unstable, but there are no significant differences in other variational bounds. This experiment shows that the network size has a small impact on the validity of this test on CIFAR-10.

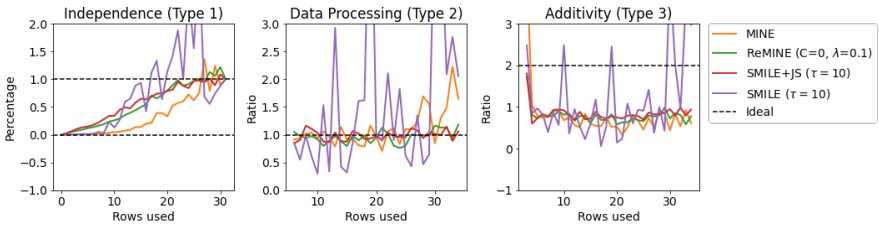

Figure 14: Self-consistency tests on CIFAR-10 with ResNet18.

**Effectiveness on the Conditional Mutual Information Estimation Task**    We compare the performance of various estimators on the conditional MI (CMI) estimation task. To set the baseline, we chose CCMI (Mukherjee et al., 2020), MINE (Belghazi et al., 2018), and KSG estimator (Kraskov et al., 2004). The Experiment is performed under the Model 1 setting in Mukherjee et al. (2020). We refer to the supplementary of Mukherjee et al. (2020) for hyper-parameter settings such as network structures and optimizer parameters. We only changed the objective function of MINE to test our method. As shown in Fig. 15, ReMINE can reach comparable performance without changing the form to classification loss. Also, ReMINE produces stable estimates compared to MINE.

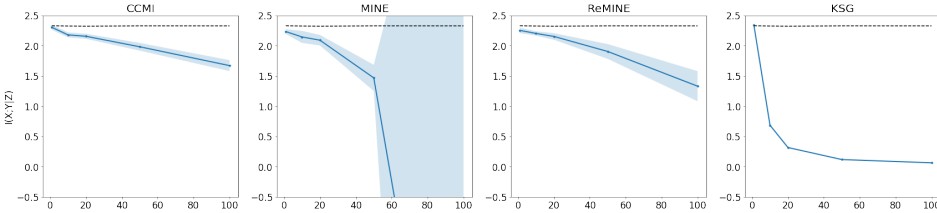

Figure 15: CMI estimation results on Model 1 ($n = 20000$). The dotted line represents the ideal CMI value. We rerun the same experiment 10 times, and plot mean (dark) and standard deviation (light).

