# OpenReview forum: "Regularized Mutual Information Neural Estimation"
_ICLR.cc/2021/Conference — Reject_

### Official Review · AnonReviewer2 · 2020-10-28
**Novel mutual information lower bound to regularized the drifting problem**

**Rating:** 5
**Confidence:** 4

**Review:**

### EDIT:

 I thank the authors for their detailed response. I also appreciate the effort that's been put into refining the draft. Unfortunately, I'm still not very happy with the motivation of attacking the drifting phenomenon on MINE. The main reason for removing the drifting effect is for moving average of history outputs. However, there are various ways for tackling this ( as pointed out in my original reviews, like using a non-drifted mutual information estimator with moving average or plugin some robust density estimators). Yes, I agree with the author the drifting phenomenon is not the only problem that the proposed method solves. But actually, the stability of other MI estimators also allows them to avoids having exploding network outputs.

 Also, in practice, people don't usually run moving average on MI for representation learning. I encourage the author to explore the importance of moving average of MI estimators further.

R3 suggests the author take some non-parametric estimators as baselines. But I think it's fine to only compare to some parametric(variational) methods on high dimension setting, where most non-parametric estimators fail. Nonetheless, it's always good to have additional experiments compared to some non-parametric methods in low dimension settings.

Overall, I lean toward rejection given current concerns.

Summary

The paper introduces a generalized version of the mutual information neural estimation (MINE), termed regularized MINE (ReMINE). Some interesting experimental results are firstly provided on a synthetic dataset: the constant term in the statistical network is drifting after MI estimate converges. Also, the optimization of MINE will result in the bimodal distribution of the outputs, in which the statistical network has very distinct values for joint and non-joint samples. The paper presents a theoretical explanation for the drifting phenomenon. In light of this, the author's approach is to add a regularized term to prevent the drifting phenomenon. They impose a $L_2$ regularization on the logsumexp term to enforce the network to find a single solution. Further, the authors make use of the historical estimation for better performance. Empirically, the proposed regularized term works well along with the original MINE estimator and ReMINE  has better performance in the continuous domain

Contributions

i) Proposal of a novel regularized MINE objective for solving the drifting phenomenon. The new objective successfully finds a single solution and exhibits a lower variance.

ii) Provide interesting insights, such as drifting phenomenon and the instability due to small batch size, out of experiments.

iii) Experimental validation of the proposed method for solving the drifting problem. Achieve better performance for mutual information estimation in the continuous domain.

Issues:

i) The drifting phenomenon of MINE is a feature but not a bug, since the drifting term has no effect on the final MINE estimated value. The motivation and benefits of solving drifting problems are unclear to me.

ii) Is the drifting phenomenon of the statistical network ubiquitous among the density ratio estimators? For example, does it exist in the density ratio estimator in logistic regression or in JS dual lower bound? If not, we can directly plug these non-drifting density estimators in MINE, instead of regularized MINE. Another apparent remedy is to make the output of statistical network $T$ zero-meaned by subtracting the online sample mean of $T$.

iii) The proposed ReMINE is motivated by the drifting phenomenon. But it can also alleviate the exploded outputs / bimodal distribution of the outputs since ReMINE explicitly imposes $L_2$ constraint on its output. The connection between the exploded outputs / bimodal distribution of the outputs and ReMINE is weak in the paper.

iv) The paper states that MINE must have a batch size proportional to the exponential of true MI to control the variance. The statement is wrong. Yes, the variance of some mutual information estimator, like NWJ,  is proportional to the exponential of true MI, as proved in [1]. However, the variance of MINE is not proportional to the exponential of true MI in finite sample case (in asymptotic maybe), due to the log function.

Minors:
a) I wonder whether the SMILE estimator (cited in the paper) implicitly solves the drifting problem. Since the optimal statistical network cannot drift freely in SMILE.

[1]A Theory of Usable Information Under Computational Constraints, Xu et al, ICLR20.

---

> ### Author Response · Authors · 2020-11-15
> **Response to Reviewer 2 (Part I)**
>
> We sincerely appreciate everyone’s comments, which were indeed helpful in improving this paper. R2’s reviews were very helpful, especially in a theoretical sense. We address the concerns raised by R2 in order of appearances.
>
> (1)  The drifting phenomenon of MINE is a feature but not a bug, since the drifting term has no effect on the final MINE estimated value. The motivation and benefits of solving drifting problems are unclear to me.
>
> To produce a single estimate with the history of network outputs, both macro- and micro-averaging strategies require the outputs to be not drifting. (Paragraph below Algorithm 1) We also provide the proof for Theorem 6 in the appendix.
> Also, removing the constant $C$ of the network output enables us to directly interpret the network output values across different batches. (Figure 4 caption)
>
> (2) Is the drifting phenomenon of the statistical network ubiquitous among the density ratio estimators? … If not, we can directly plug these non-drifting density estimators in MINE, instead of ReMINE.
>
> As R1 suggested, other bounds such as JS or MINE-$f$ might be used to avoid the drifting problem. However, the drifting phenomenon is not the only problem that our method solves. Three bullet points in Section 3 shows the advantages of our method:
> * Solves the drifting phenomenon: Section 4
> * Avoids having exploding network outputs: By implicitly adjusting the gradient step size by using L2 regularization. (Section 5)
> * Interpretable network outputs: “We can now directly interpret the network outputs, and $E(j)$ directly converges to ideal MI $\log N$, thanks to the regularization of $C$.” (Section 5 Figure 4)
>
> (3) Is the drifting phenomenon of the statistical network ubiquitous among the density ratio estimators? … Another apparent remedy is to make the output of statistical network $T$ zero-meaned by subtracting the online sample mean of $T$.
>
> Note that for the optimal $T$, first term $E_{P}[T] \to I(X; Y)+ C$ and second term $\log(E_{Q}[e^T])  \to C$. There are two apparent problems when subtracting the online sample mean.
>
> First, as seen in Figure 3 (d), the second term output tends to get quite noisy. The reason behind it is in Subsection 3.2 “Exploding network outputs”. The blue line in Figure 3 (d) can be viewed as the output of the online sample mean strategy, which fails to be stabilized. Also in Figure 6, the light blue line is the estimate of each batch. In other words, getting a stable online estimate of C from a single batch is hard. This naturally leads to unstable gradients, which can be another answer to the above question (2).
>
> Secondly, C changes for every batch. This is why the original MINE also had to use heuristics like the exponential moving average (Subsection 3.2 from Belghazi et al., 2018) to avoid the inherent bias generated by the averaging strategies (Theorem 6). Hence, the exponential moving average used in other literature such as (Poole et al., 2019) has limitations, as seen in Figure 1(c). Note that the red line is the exponential moving average, and the axis for figure 1(c) is logarithmic.
> Besides, since we constrained the estimate of the second term in DV to have the same value for each batch, we relaxed the constraint on sample complexity from a single batch to the number of samples in the history (Algorithm 1).

---

> ### Author Response · Authors · 2020-11-15
> **Response to Reviewer 2 (Part II)**
>
> (4) The proposed ReMINE is motivated by the drifting phenomenon. But it can also alleviate bimodal distribution of the outputs since ReMINE explicitly imposes a constraint on its output. The connection between the exploded outputs / bimodal distribution of the outputs and ReMINE is weak in the paper.
>
> Exploding outputs:
> We analyze that ReMINE mitigates the exploding outputs in the gradient descent-based optimization by implicitly controlling the step size of the gradient (Subsection 5.1).
>
> Bimodal distribution of the outputs:
> First of all, as the statistics network of ReMINE converges to $\log \frac{dp}{dq}$ (log-likelihood ratio), it is natural to be having bimodal outputs. ($\frac{dp}{dq}=1$ for joint cases, and $\frac{dp}{dq}=0$ for non-joint cases, as our toy dataset is discrete.) (see Theorem 5.) So, we don’t think that ReMINE suppresses the bimodal distribution of the outputs, at least in a theoretical sense.
>
> However, in a gradient-descent perspective, large lambda (regularizer weight) is indeed not helpful, as the non-joint sample network outputs converge to $-\frac{1}{2 \lambda}$ in the discrete case. (Explanation in the subsection 5.1, and Figure 4(b) also shows that non-joint sample network outputs converge to $-\frac{1}{2 * 0.1} = -5$)
> We additionally show the performance of ReMINE on our toy dataset in a range of lambda in Figure 8(c). Empirically, we found that using values near 0.1~1.0 helped performance. We used 0.1 or 1.0 for all our experiments except Figure 8(c).
>
> Additionally, without removing $C$ with ReMINE, we believe that we shouldn’t directly interpret the network output values, but it is interpreted as a relative size. We believe Fig 2(b) and Fig 4(b) will also clarify this.
>
> (5) The paper states that MINE must have a batch size proportional to the exponential of true MI to control the variance. The statement is wrong. Yes, the variance of some mutual information estimator, like NWJ, is proportional to the exponential of true MI, as proved in [1]. However, the variance of MINE is not proportional to the exponential of true MI in finite sample cases (in asymptotic maybe) due to the log function.
>
> By reflecting R2's comment, the sentence was revised like this:
> “where the batch size correlates with the variance of the MI estimates.”
>
> (6) I wonder whether the SMILE estimator (Song & Ermon, 2020) implicitly solves the drifting problem since the optimal statistical network cannot drift freely in SMILE.
>
> We found that SMILE (without JS) behaves quite similarly with MINE, as the gradient gets clipped off whenever network outputs are not inside the fixed bound.
>
>
> (Belghazi et al., 2018): Mohamed Ishmael Belghazi, Aristide Baratin, Sai Rajeshwar, Sherjil Ozair, Yoshua Bengio, Aaron Courville, and Devon Hjelm. Mutual information neural estimation. In International Conference on Machine Learning, pp. 531–540, 2018.
> (Poole et al., 2019): Ben Poole, Sherjil Ozair, Aaron Van Den Oord, Alex Alemi, and George Tucker. On variational bounds of mutual information. In International Conference on Machine Learning, pp. 5171–5180, 2019.
> (Song & Ermon, 2020): Jiaming Song and Stefano Ermon.  Understanding the limitations of variational mutual information estimators. In International Conference on Learning Representations, 2020.  URL https://openreview.net/forum?id=B1x62TNtDS.

---

### Official Review · AnonReviewer1 · 2020-10-29
**Good paper with new insights for the community**

**Rating:** 7
**Confidence:** 2

**Review:**

This paper attempts to answer the four questions raised from the mutual information estimator. To this end, this paper investigates why the MINE succeeds or fails during the optimization on a synthetic dataset. Based on the observations and discussions, the paper then proposes a novel lower bound to regularize the neural networks and alleviate the problems of MINE.

Overall, the paper is easy to follow and new insights have been brought for the MI estimator and the downstream tasks.

---

> ### Author Response · Authors · 2020-11-15
> **Response to Reviewer 1**
>
> We thank R1 for recognizing our efforts made in this work.

---

### Official Review · AnonReviewer4 · 2020-10-29
**Regularized MINE review**

**Rating:** 6
**Confidence:** 3

**Review:**

The work studies a neural-network based estimator, referred to as MINE, for approximating the mutual information between two variables.  By designing a synthetic dataset, the work studies properties of MINE and, based on these findings, proposes a new method incorporating regularization, called ReMINE, that they empirically demonstrate has nice performance.

Strengths:
-- The proposed ReMINE algorithm combats drifting of the two terms in MINE's approximation of the DV representation through regularization that forces the network to find a single solution (as opposed to a family of solutions).
-- The proposed method is novel and performs well compared to state-of-the-art.

Weaknesses:
-- I found the writing to be difficult to understand and follow. For example, the paper is not self-contained: the authors use terms like "statistical network" without providing definitions, and begin simulations without introducing MINE or giving a high-level discussion of how it estimates the DV representation, so it would be very difficult to read this paper without having first read the original MINE paper.
-- The paper lacks intuition. For example, there is little to no discussion of why the synthetic dataset proposed is a good choice for studying the underlying properties of MINE.

Some additional typos/comments:

Pg 1: "...the value of each term in MINE loss IS shifting even..."
Pg 2: I believe equation (3) should be a lower bound not an equality.
Pg 2: "...where estimates of  __ and __ DRIFT in parallel..."
Pg 4: mu and sigma in Figure 2 were never defined.
When printed, the black-and-white figures are too small to read and interpret.

---

> ### Author Response · Authors · 2020-11-15
> **Response to Reviewer 4**
>
> We sincerely appreciate everyone’s comments, which were indeed helpful on improving this paper. R4’s detailed review helped improve the readability of this paper. We address the concerns raised by R4 in order of appearances.
>
> (1) I found the writing to be difficult to understand and follow. For example, the paper is not self-contained: the authors use terms like "statistical network" without providing definitions, and begin simulations without introducing MINE or giving a high-level discussion of how it estimates the DV representation, so it would be very difficult to read this paper without having first read the original MINE paper.
>
> To improve the readability of our paper, we added the definition of the statistic network in Section 2, paragraph “Variational Mutual Information Estimation”. We also added another section “Appendix: Comparison with Mutual Information Neural Estimator” in the appendix.
>
> (2) The paper lacks intuition. For example, there is little to no discussion of why the synthetic dataset proposed is a good choice for studying the underlying properties of MINE.
>
> The ultimate reason behind the synthetic dataset is “easily discerning samples of joint distribution from marginal distribution (Subsection 3.1 “Dataset”)”. Without this, we cannot visualize Figure 1 and 2, and clearly show the drifting problem.
>
> (3) Some additional typos/comments:
>
> All comments are reflected in the revised version except the figure. We are going to re-draw all the graphs in the camera-ready phase. Thank you for your careful review.

---

### Official Review · AnonReviewer3 · 2020-11-03
**Major revisions needed**

**Rating:** 3
**Confidence:** 5

**Review:**

Response to authors: After reading the authors' response, I have decided to maintain my original rating. The authors have not adequately addressed my main concerns.

Novelty: The work here, as indicated by the authors, is largely an incremental improvement over an existing work MINE. The authors' response did not alleviate this concern and in fact reinforced it.

Citations and comparisons to other work: The authors did not agree to even include citations to important literature in this area. This should have been a bare minimum and it is a mistake for variational approaches to ignore these works which have theoretical guarantees that many variational approaches do not have. Comparisons to other methods should also have been included. The methods the authors did compare to have weak (or no) theoretical guarantees for higher dimensions.

Theoretical work: The authors simply pointed to the theoretical work for MINE. However, the theoretical work in MINE is also very weak and only focuses on estimation consistency and not convergence rates (i.e. the statistical bias and variance of the estimator). More theoretical work is needed in this area to justify the use of these estimators over others.

Some responses to other comments that may help the authors with further revisions:

(1) The presentation of MINE should occur in the main paper as this is crucial for understanding the paper.

(2) This was not clear. Perhaps the authors could include similar pointers in the paper with each of these issues.

(3) The bias I'm referring to here is the actual statistical bias of the estimator. From Theorem 6, it seems that the drift problem does seem to create some bias but it would be useful to quantify that, which could then lead to a bias correction approach.

(7) The way this is currently worded, it sounds like you are saying that training with a larger batch size is bad. This part should be clarified to avoid this.

Original Review:

This paper presents a modified version of a neural network-based MI estimator. They investigate a few of the issues of this specific estimator and propose a regularization to help with one of them. MI estimation is an important and difficult topic. Improvements in this area are of definite interest.

Pros:
The paper appears to be technically correct. The experiments are somewhat supportive of including the regularization, especially when the MI is higher which is a known issue with some MI estimators.

Cons:
There are some interesting ideas here but the paper feels unpolished. The presentation of the ideas is somewhat unconventional. Several issues with the MINE estimator are presented and then two of them are discarded in favor of a focus on one of them. The paper could benefit from a bit more focus in this regard. In the end, the authors really only propose a small modification to the MINE estimator to counter the supposed drifting problem and do some experiments showing some improvement. But it's not clear how much of a problem this drift really is. The authors show that it causes a bias but they do not present how much bias it adds.

In addition, the authors are severely neglecting some of the non-neural network state of the art MI estimators in their citations and comparisons (see the references below for some examples, which all have strong theoretical results).

The theoretical work is also weak with regards to the convergence rates of the proposed estimator as well. While empirical results can confirm that an estimator can be useful in practice, they are easy to cherry-pick and ultimately theoretical guarantees are needed to know an estimator's general performance. Thus the results would be a lot stronger if convergence rate guarantees were given.

Other comments/questions:
The authors should define the MINE estimator in this paper.

The second bullet point on page 2 says that "training with larger batch size reduces the variance of the MI estimate". Isn't this a good thing? That would lead to better convergence.

In Section 3.1 the notation is technically incorrect. Instead of stating $I(X;X)$ it should be written as $I(X_1;X_2)$ where $X_1$ and $X_2$ are i.i.d. The former suggests that you're comparing the same random variables.

On page 4, it's suggested that joint samples are sparse with reduced sample size. Why aren't joint samples simply included together during training?

Does regular L2 regularization help with the drift problem?

[R1] Moon et al."Ensemble estimation of mutual information," ISIT, 2017.
[R2] Moon et al., "Information theoretic structure learning with confidence," ICASSP, 2017.
[R3] Moon et al., "Ensemble Estimation of Information Divergence," Entropy, 2018.
[R4] Singh and Poczos, "Exponential concentration of a density functional estimator," NeurIPS, 2014.
[R5] Kandasamy et al., "Nonparametric von Mises estimators for entropies, divergences, and mutual informations," NeurIPS. 2015.

---

> ### Author Response · Authors · 2020-11-15
> **Response to Reviewer 3 (Part I)**
>
> We sincerely appreciate everyone’s comments, which were indeed helpful in improving this paper. Especially, we added another section based on R3’s comments, which made this paper more readable. We address the concerns raised by R3 in order of appearances.
>
> (1) The presentation of the ideas is somewhat unconventional.
>
> As both R3 and R4 have noted, our paper did not provide the explanations on MINE clearly enough. To address the concerns, we added another section “Appendix: Comparison with Mutual Information Neural Estimator” in the appendix. We hope the added section eases the readability of the paper.
>
> (2) Several issues with the MINE estimator are presented and then two of them are discarded in favor of a focus on one of them.
>
> We did not discard any issues. In this work, we raise 4 issues in total (Bullet points in the Introduction):
>
> Q. How does the neural network inside MINE behave when estimating MI?
> A. We analyze the network indirectly by analyzing the network outputs, as shown in Section 3.2, Figure 1, and 2.
>
> Q. Why does MINE loss diverge in some cases? (oscillates, and converges to infinity,  etc.) Where does the instability originate from?
> A. “Exploding network outputs” in subsection 3.2 points out the problematic gradients in MINE, and the first paragraph in subsection 5.1 “Effectiveness of L2 regularization” analyzes the gradient of ReMINE, which helps avoiding the output explosion problem.
>
> Q. Can we make a more stable estimate on small batch size settings?
> A. We point out the drifting problem in Figure 1 (a), first paragraph in 3.2 “Observation”, Section 4, and we propose a novel strategy (micro-averaging strategy) in Algorithm 1, which cannot be used when the drifting problem occurs (Theorem 6).
>
> Q. Why does the value of each term in MINE loss are shifting even after the estimated MI
> converges? Are there any side effects of this phenomenon?
> A. Same as above. (Section 4) Hence MINE can only rely on single batch estimates (Paragraph after Algorithm 1).
>
> (3) It's not clear how much of a problem this drift really is. The authors show that it causes a bias but they do not present how much bias it adds.
>
> We did not fully understand what “bias” the R1 was pointing out. Note that the drifting phenomenon does not cause a biased MI estimate for a single-batch estimate.
>
> 1) For the bias in the estimated MI from a single batch:
> Figure 1 (a) experimentally shows the drifting, and Section 4 is dedicated to understanding the phenomena. However, MI estimate for the single batch is not biased, as shown in Figure 1 (a) and proven in the DV representation.
>
> 2) For the bias in the estimated MI from multiple batches:
> Theorem 6 shows that both macro- and micro-averaging strategies produce a biased estimate when the drifting problem occurs. (e.g. Averaging the batchwise estimate of MI leads to a biased estimate.)
>
> 3) For the bias in the gradient estimates:
> MINE tries to mitigate the gradient bias by exponential moving average of mini-batches. (Paragraph below Lemma 1, Appendix: Comparison with Mutual Information Neural Estimator) ReMINE does not solve the biased gradient problem, but it mitigates the gradient explosion problem by implicitly controlling the step size of the gradient (Subsection 5.1).
>
> (4) In addition, the authors are severely neglecting some of the non-neural network state of the art MI estimators in their citations and comparisons.
>
> Similar to (Poole et al., 2019), we concentrate on the variational bounds of mutual information so that it can be used on representation learning or generative models (as noted in the first paragraph of the introduction). We believe that our paper is an incremental study from MINE, hence fair to compare with neural-network based MI estimators with our method. Nonetheless, we compared our method with non-neural network MI estimators such as (Jiao et al., 2018) and (Gao et al., 2017) which has strong theoretical guarantees with the benchmark experiment provided in (Poole et al., 2019) in the appendix.
>
> (5) The theoretical work is also weak with regards to the convergence rates of the proposed estimator … The authors should define the MINE estimator in this paper.
>
> Convergence proofs are already provided in the MINE paper, and we wrongly thought the extension was obvious. To address the reviewers’ concerns on the issue of self-containment of our paper (R2, R3), we add “Appendix: Comparison with Mutual Information Neural Estimator” in the appendix.

---

> ### Author Response · Authors · 2020-11-15
> **Response to Reviewer 3 (Part II)**
>
> (6) While empirical results can confirm that an estimator can be useful in practice, they are easy to cherry-pick …
>
> As it can be seen in the provided source code, we simply added our ReMINE implementation to the existing code base of (Poole et al., 2019). We did not modify the provided benchmark experiments.
> For experiments of SMILE (Self-consistency test, Song & Ermon, 2020), we couldn’t find the relevant source code except for the implementation of the SMILE loss itself. We re-implemented the experiment (which is also included in the provided source code) and reproduced the similar results on SMILE, InfoNCE, and JS with 10 repeated runs.
> For experiments of CMI (Mukherjee et al., 2020), we also conduct 10 repeated runs from the source code the authors provided. We added our implementation to the provided code.
>
> (7) The second bullet point on page 2 says that "training with larger batch size reduces the variance of the MI estimate." Isn't this a good thing? That would lead to better convergence.
>
> Yes, it is a good thing. Following the work of (McAllester & Stratos, 2018) and (Song & Ermon, 2020), we experimentally observe the relationship between the batch size and the estimation variance.
> However, the problems are when batch size is small: “Exploding network outputs, where smaller batch sizes cause the network outputs to explode.”
>
> (8) In Section 3.1, the notation is technically incorrect. ... The former suggests that you're comparing the same random variables.
>
> As you have pointed out, it is not i.i.d., rather, we are indeed comparing exactly the same random variable. Namely, $\frac{dP}{dQ} = 1$  for joint samples and $\frac{dP}{dQ} = 0$ for non-joint samples.
>
> (9) On page 4, it's suggested that joint samples are sparse with reduced sample size. Why aren't joint samples simply included together during training?
>
> Because it will change the distribution of $Q$ (namely, to $Q’$), hence the optimal network converges to the different $\frac{dP}{dQ}$ (namely, to $\frac{dP}{dQ'}$). Also, this gets much harder to provide a method that works in a general sense if samples weren’t discrete (e.g. Experiment shown in Figure 6 and Figure 9).
>
> (10) Does regular L2 regularization help with the drift problem?
>
> Yes. We provided proofs (Section 4), experiments (Figure 4, 5), analysis on the gradients (Subsection 5.1) and the experimental results using L1 regularization (Figure 11).
>
> (Poole et al., 2019): Ben Poole, Sherjil Ozair, Aaron Van Den Oord, Alex Alemi, and George Tucker. On variational bounds of mutual information. In International Conference on Machine Learning, pp. 5171–5180, 2019.
> (Jiao et al., 2018): Jiantao Jiao, Weihao Gao, and Yanjun Han. The nearest neighbor information estimator is adaptively near minimax rate-optimal. In Advances in neural information processing systems,  pp. 3156–3167, 2018.
> (Gao et al., 2017): Weihao Gao, Sreeram Kannan, Sewoong Oh, and Pramod Viswanath.  Estimating mutual information for discrete-continuous mixtures. In Advances in neural information processing systems, pp.5986–5997, 2017
> (Song & Ermon, 2020): Jiaming Song and Stefano Ermon.  Understanding the limitations of variational mutual information estimators. In International Conference on Learning Representations, 2020.  URL https://openreview.net/forum?id=B1x62TNtDS.
> (Mukherjee et al., 2020): Sudipto Mukherjee, Himanshu Asnani, and Sreeram Kannan.  Ccmi:  Classifier based conditional mutual information estimation.  In Uncertainty in Artificial Intelligence, pp. 1083–1093. PMLR, 2020.
> (McAllester & Stratos, 2018): David McAllester and Karl Stratos.  Formal limitations on the measurement of mutual information. arXiv preprint arXiv:1811.04251, 2018.

---

### Decision · Program_Chairs · 2021-01-07
**Final Decision**

**Decision:**

Reject

**Comment:**

This paper is a study in optimizing the Donsker-Varadhan lower bound on mutual information focusing on a "drift" problem.  The bound is a difference between terms which appears to have an extra degree of freedom where the two terms increase or decrease together.  They propose a fix for this problem. The authors state that the DV bound is of practical value but in most cases it is replaced by discriminative lower bounds as in contrastive predictive coding (CPC) which are biased but have lower variance. The paper does not address the variance (convergence) issues with the DV bound.

We have a well informed reviewer who feels that the paper is not sufficiently novel and has other issues supporting rejection.  Other reviews are not very enthusiastic.  I will side with rejection.